# Noisy Computing of the Threshold Function

**Ziao Wang**                                                                    ZIAOW@UMICH.EDU
*Department of Electrical Engineering and Computer Science, University of Michigan, Ann Arbor*

**Nadim Ghaddar**                                                   NADIM.GHADDAR@UTORONTO.CA
*Department of Electrical and Computer Engineering, University of Toronto*

**Banghua Zhu**                                                                  BANGHUA@UW.EDU
*Department of Electrical and Computer Engineering, University of Washington*

**Lele Wang**                                                               LELEWANG@ECE.UBC.CA
*Department of Electrical and Computer Engineering, University of British Columbia*

**Editors:** Gautam Kamath and Po-Ling Loh

## Abstract

Coping with noise in computing is an important problem to consider in large systems. With applications in fault tolerance (Hastad et al., 1987; Pease et al., 1980; Pippenger et al., 1991), noisy sorting (Shah and Wainwright, 2018; Agarwal et al., 2017; Falahatgar et al., 2017; Heckel et al., 2019; Wang et al., 2024a; Gu and Xu, 2023; Kunisky et al., 2024), noisy searching (Berlekamp, 1964; Horstein, 1963; Burnashev and Zigangirov, 1974; Pelc, 1989; Karp and Kleinberg, 2007), among many others, the goal is to devise algorithms with the minimum number of queries that are robust enough to detect and correct the errors that can happen during the computation.

In this work, we consider the noisy computing of the threshold-$k$ function. For $n$ Boolean variables $\mathbf{x} = (x_1, \ldots, x_n) \in \{0,1\}^n$, the threshold-$k$ function $\mathsf{TH}_k(\cdot)$ computes whether the number of 1's in $\mathbf{x}$ is at least $k$ or not, i.e.,

$$\mathsf{TH}_k(\mathbf{x}) \triangleq \begin{cases} 1 & \text{if } \sum_{i=1}^n x_i \geq k; \\ 0 & \text{if } \sum_{i=1}^n x_i < k. \end{cases}$$

The noisy queries correspond to noisy readings of the bits, where at each time step, the agent queries one of the bits, and with probability $p$, the wrong value of the bit is returned. It is assumed that the constant $p \in (0, 1/2)$ is known to the agent. Our goal is to characterize the optimal query complexity for computing the $\mathsf{TH}_k$ function with error probability at most $\delta$.

This model for noisy computation of the $\mathsf{TH}_k$ function has been studied by Feige et al. (1994), where the order of the optimal query complexity is established; however, the exact tight characterization of the optimal number of queries is still open. In this paper, our main contribution is tightening this gap by providing new upper and lower bounds for the computation of the $\mathsf{TH}_k$ function, which simultaneously improve the existing upper and lower bounds.

The main result of this paper can be stated as follows: for any $1 \leq k \leq n$, there exists an algorithm that computes the $\mathsf{TH}_k$ function with an error probability at most $\delta = o(1)$, and the algorithm uses at most

$$(1 + o(1)) \frac{n \log \frac{m}{\delta}}{D_{\mathsf{KL}}(p \| 1 - p)}$$

---

Extended abstract. Full version appears as (Wang et al., 2024b).

This work was completed while Ziao Wang was with the University of British Columbia.

queries in expectation. Here we define $m \triangleq \min\{k, n - k + 1\}$ and denote the Kullback-Leibler divergence between $\mathsf{Bern}(p)$ and $\mathsf{Bern}(1-p)$ distributions by $D_{\mathsf{KL}}(p\|1-p)$. Conversely, we prove that to achieve an error probability of $\delta = o(1)$, any algorithm must make at least

$$(1 - o(1))\frac{(n - m)\log\frac{m}{\delta}}{D_{\mathsf{KL}}(p\|1 - p)}$$

queries in expectation. When $m = o(n)$, the ratio between these upper and lower bounds is $1 + o(1)$, and hence we provide an asymptotically tight characterization for the optimal number of queries. For general $m$, these bounds are tight within a multiplicative factor of 2. When specialized to the case of $k = 1$, our results recover the optimal bounds for computing the logic OR function found by Zhu et al. (2024).

**Keywords:** Noisy computing, sample complexity, active learning

## Acknowledgement

This work was supported in part by the Natural Sciences and Engineering Research Council of Canada (NSERC) Discovery under Grant No. RGPIN-2019-05448, in part by the NSERC Collaborative Research and Development under Grant CRDPJ 54367619, and in part by the National Science Foundation under Grants IIS-1901252 and CCF-1909499.

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
