# OpenReview forum: "Noisy Computing of the Threshold Function"
_algorithmiclearningtheory.org/ALT/2025/Conference — ALT 2025_

### Official Review · Reviewer_BbYK · 2024-10-20
**Review for submission 78**

**Rating:** 8
**Confidence:** 4

**Review:**

I was involved in the reviewing of a previous iteration of this paper. There were correctness concerns before, which (as far as I can tell) have now been mitigated in the current submission.

This paper is concerned with the evaluation of the $k$-out-of-$n$ threshold function, under the model where each input bit can only be queried noisily. Specifically, the model assumes that each input bit query is i.i.d., and modeled by a Bernoulli coin with bias $p$, with the same bias $p$ across all the bits. The paper gives matching upper and lower bounds up to a $1+o(1)$ factor in query complexity in the regime where both $k = o(n)$ (and symmetrically $n-k = o(n)$) and $\delta = o_n(1)$, and bounds that match up to a factor of 2 for general $k$.

Both the upper and lower bounds are tackled by case analyses depending on the value of $k$ (though the regime splits are different for the upper and lower bounds). The lower bound was proven by a combination of the two-point method and a genie-aided-algorithm argument. The upper bound was given by two algorithms in two different regimes. One minor comment is that I wish the authors would explain why the regime split in the upper bound is exactly at that point. Currently the algorithm descriptions in Section 4 are a little bit prescriptive and not as well-motivated as the lower bound.

Overall, I think the paper uses an interesting combination of techniques (even if many are existing techniques, but the uses are interesting, e.g. the MaxHeapThreshold use after filtering a lot of bits), and the result itself is significant in that it tackles and yields substantial progress on the sharp constant question. Given that (for me) there are no longer any correctness concerns, I advocate for accepting the paper.

**Paper Award:**

No

---

> ### Author Response · Authors · 2024-11-24
>
> We thank the reviewer for the valuable comments and feedback. We also deeply appreciate the opportunity to engage in multiple rounds of discussion with the reviewer during the previous iteration of our manuscript submission, which significantly enhanced the technical rigor and overall presentation of our paper.
>
> Regarding the reviewer’s comment on the case split in the upper bound, we would like to provide some intuition on why Algorithm 1 performs better for the larger $k$ regime, while Algorithm 2 works better for the smaller $k$ regime.
>
> First, let's suppose $k = n/2$, and for simplicity, consider the challenging case where the input $x$ contains either $k$ ones or $k - 1$ ones. We can think about this as having a (roughly) uniform prior on each bit, and each bit is equivalently important in determining the evaluation of the $TH_k$ function. This justifies why it is desirable to spend the same expected number of queries in determining the value of each bit, as Algorithm 1 does.
>
> On the other hand, when $k$ is small, such as $k = \sqrt{n}$, the situation differs significantly. Consider again the case where the input contains either $k$ ones or $k - 1$ ones. Since a dominating fraction of the bits are zeros, a bit of value 1 carries much more information in determining the $TH_k$ function compared to a bit of value zero. As a result, Algorithm 1, which spends the same expected number of queries on each bit, becomes inefficient. Instead, a natural approach is to filter out a dominating fraction of bits that are “apparently” zeros and focus on the rest of the bits to determine the $TH_k$ function. This strategy underpins the design of Algorithm 2.
>
> Finally, we would like to point out that $n / \log n$ is not the only choice for the split point. Our analysis in Section 4 shows that to attain the desired upper bound, our usage of Algorithm 1 requires that $\log(n / \delta) = (1 + o(1)) \log(k / \delta)$, while our usage of Algorithm 2 requires that $k = o(n)$. This suggests that we can set the split point to be $n / r$, where $r$ is any function of $n$ that satisfies $r = \omega(1)$ and $r = n^{o(1)}$. The proof remains valid for any such choice of a split point.

---

### Official Review · Reviewer_YbKM · 2024-10-24
**recommend accept**

**Rating:** 8
**Confidence:** 5

**Review:**

This paper proves lower and upper bounds on the number of noisy queries needed to compute the threshold function (given $n$ input bits, whether at least $k$ of them are $1$s). Previously, it is known that $\Theta(n\log(k/\delta))$ queries is both necessary and sufficient (where $\delta$ is error probability), and this paper provides bounds on the leading coefficient. When $k=o(n)$, the lower and upper bounds match. For general $k$, there is a multiplicative gap of $2$ between the lower and upper bounds. Also note that the case $k=1$ is resolved in a previous work (Zhu et al. (2024)).

The lower bound (Theorem 1) occupies majority of the paper. The proof is divided into two regimes: the case $\log k \le \frac{\log (1/\delta)}{\log \log (1/\delta)}$ is proved by an easy application of Le Cam's two point method. The case $\log k > \frac{\log (1/\delta)}{\log \log (1/\delta)}$ is more difficult and is proved using a refined version of Feige et al. (1994)'s method, which reduces the query model to a two-phase version, where the first phase has non-adaptive noisy queries and the second phase has adaptive exact queries. (This second case is further divided into two sub-cases.)

The upper bound (Theorem 2) is by a simple algorithm.

In recent years there is an increased interest in determining the leading coefficient in the query complexity of noisy computation problems. The threshold problem is a very basic problem which everyone can appreciate, so it is natural problem to study. This paper is a solid step towards fully determine the leading coefficient and resolves the problem in the regime $k=o(n)$. The proof technique is not entirely novel (essentially a more detailed analysis of Feige et al. (1994)'s proof) but the work put into actually performing this more detailed analysis is non-trivial.

The paper is well-written and easy to understand.

In summary, this paper proves nice and interesting results on a natural problem. I recommend it to be accepted.

**Paper Award:**

No

---

> ### Author Response · Authors · 2024-11-24
>
> We thank the reviewer for the valuable comments!

---

### Official Review · Reviewer_Zovh · 2024-11-09
**Noisy Computing of the Threshold Function**

**Rating:** 7
**Confidence:** 4

**Review:**

This paper studies the sample complexity of threshold functions in the context of noisy queries each of which errs with a certain probability i.i.d. The authors present both a lower bound and an upper bound on the sample complexity, which match up to a constant factor. The employed algorithms built on several pre-existing ideas, including the 'checkbit' from reference GX 23 for k≤n/log and 'maxheapthreshold' from Feige et al. 1994 for k≥n/log. The established lower bound merges ideas from the Le Cam two-point method and concepts from Feige et al. 1994.

The problem of learning threshold functions with noisy queries is a fundamental problem in learning theory. Hence, obtaining an accurate understanding of its sample complexity is of significant theoretical interest. The primary strength of this paper lies in its resolution of a fundamental theoretical problem in learning theory. However, its weakness stems from the lack of novelty in the techniques used. The authors essentially combine several existing standard techniques and perform a dedicated analysis, which, while demanding considerable effort and sophistication.
Given the fundamental nature and theoretical interest of the results, I recommend that the paper to be accepted by ALT.

**Paper Award:**

No

---

> ### Author Response · Authors · 2024-11-24
>
> We thank the reviewer for the valuable comments!

---

### Meta-Review · Area_Chair_yf4F · 2024-12-05

**Recommendation:** Accept
**Confidence:** 4

**Metareview:**

This paper gives precise upper and lower bounds on the number of noisy queries required to compute threshold functions.  The reviewers are positive about all aspects of the paper, including the contributions, correctness, and clarity.  It was noted that the technical novelty might not be among the highest of submissions given the similarity to previous literature on noisy algorithms, but even so, the findings are by no means straightforward given existing works.  Overall, the decision is clear acceptance.

**Paper Award:**

No